# Altitudinal Visual Field Defects Following Diagnostic Transfemoral Cerebral Angiography

**DOI:** 10.3390/medicina57060567

**Published:** 2021-06-03

**Authors:** Kyungwoo Yoon, Soo Young Chae, Kiyoung Kim

**Affiliations:** 1Department of Ophthalmology, Kyung Hee University Hospital, Kyung Hee University, Seoul 02447, Korea; sky00forever@daum.net; 2Departments of Radiology, Seoul Saint Mary’s Hospital, College of Medicine, The Catholic University of Korea, Seoul 02447, Korea; csyloo@naver.com

**Keywords:** transfemoral cerebral angiography, intracranial aneurysm, ischemic optic neuropathy, visual field defect

## Abstract

Transfemoral cerebral angiography (TFCA) has been increasingly used as diagnostic method for the evaluation of cerebral vessels. Ophthalmologic complication after TFCA has rarely been reported, and most complications are associated with an intraoperative thrombo-embolic event. We reported a patient who developed a superior altitudinal visual field defect one day after diagnostic TFCA. The ophthalmic exam revealed a prominent inferior optic disc edema, and the fluorescein angiographic showed the non-perfusion of the corresponding inferior sectoral optic disc. Diffusion-weighted MRI on the day following cerebral angiography revealed multiple focal scattered acute infarctions. Even one month after steroid pulse therapy, the superior nasal field defect remained with minimal improvement. We believe this case was consistent with an acute anterior ischemic optic neuropathy (AION) due to thrombo-embolism after TFCA. Ophthalmic examinations and a visual field test should be performed before and immediately after the TFCA, particularly in the case with a high risk of thromboembolic events.

## 1. Introduction

Cerebral digital subtraction angiography (DSA) is an endovascular diagnostic procedure used to evaluate intracranial aneurysm, subarachnoid hemorrhage (SAH), stroke, and arteriovenous malformations. The incidence of neurological complications following cerebral DSA occurred in 2.63%, especially ophthalmic complications, which constituted 0.24% [1]. The reasons for these complications are not completely understood; however, intraoperative ischemic events can be caused by microemboli, which are confirmed by lesions on diffusion-weighted MRI (DWI) or intraoperative transcranial Doppler [2]. It was previously reported that the incidence of transient complications including visual field cut following cerebral DSA was 0.2%. According to another paper, permanent neuro-ophthalmological complications after cerebral angiography using DSA have been reported much rarer, about 0.02% [1,3]. A recent study found new visual defects in 33% of patients treated for ophthalmic segment internal carotid artery (ICA) aneurysms, although the majority of the population studied had undergone surgical clipping [4]. Here, we report a patient who developed altitudinal visual field defects following the DSA performed for the follow-up of previously treated aneurysms.

## 2. Case Presentation

A 59-year-old woman underwent transfemoral cerebral angiography (TFCA) for a follow-up evaluation of previously treated intracranial aneurysms. She had a history of SAH secondary to the left posterior communicating artery (PCOM) aneurysm that was treated with coil embolization 7 years prior. One year later, she underwent the surgical clipping of a right PCOM aneurysm and fully recovered without any neurological deficit. In brain computer tomography angiography, before TFCA, there was no definite chronic degenerative changes or focal lesion in the brain parenchyma. On TFCA, the blood flow of the ophthalmic artery was normal, and the previously clipped aneurysm was noted on the right PCOM (Figure 1A). Additionally, an intact choroidal blush was observed during the late arterial phase (Figure 1B). The morning after TFCA, she complained of visual disturbance of her superior visual field. The patient presented no headaches, diplopia, or jaw claudication. Ophthalmic examination revealed that her visual acuity was 20/20 in the right eye. There was no limitation or pain with extraocular muscle movement, and the relative afferent pupillary defect (RAPD) was negative. She promptly underwent brain DWI and it showed multiple focal scattered acute infarctions, possibly due to embolic events (Figure 1C,D).

Fundus examinations revealed inferior sectoral optic disc swelling and small cotton wool spots (CWS) in the peripapillary area of the right eye. Fluorescein angiography (FA) showed early filling defects in the inferior segment of the optic nerve’s head and late leakage from the optic disc. A Humphrey visual field 24-2 showed a superior altitudinal deficit that corresponds with inferior sectoral papilledema (Figure 2). We assumed the diagnosis of acute nonarteritic anterior ischemic optic neuropathy (NAION) caused by posterior ciliary artery occlusion from micro-embolism after TFCA. Orbit MRI was also performed to exclude optic neuritis, and optic nerve was not enhanced. Carotid ultrasonography revealed mild stenosis on the right distal common carotid artery (22%) and right carotid bulb (28%). Two days after onset, prednisolone (1000 mg/day) was intravenously administered for 3 days as steroid pulse therapy. Papilledema had gradually improved except for the inferior temporal sector, but the visual field defect remained with no significant change up to 7 days after the start of the treatment. One month later, most papilledema and peripapillary CWS were resolved on fundus photography. However, narrowed inferior neuroretinal rim and peripapillary retinal nerve fiber layer (RNFL) thinning was observed in the corresponding quadrant on SD-OCT (Figure 3). Although the visual field slightly improved, superior nasal quadrantanopia still remained.

## 3. Discussion

One of the largest retrospective studies reported that neurological complications occurred in 2.63% of cases after diagnostic cerebral catheter angiography [1]. Factors independently associated with an increased risk of neurological complications included the indication of atherosclerotic cerebrovascular disease, subarachnoid hemorrhage, and the comorbidity of frequent transient ischemic attacks [1]. Until now, 1.1% of ocular complications such as visual deficits have been reported, even after the successful coil embolization of carotid ophthalmic aneurysms [4]. This is the first case, to our knowledge, of acute onset quadrant visual field defect after diagnostic TFCA. This patient had a history of clipping of right PCOM aneurysm, which was not directly connected to the ophthalmic artery origin. We can thus hypothesize that microemboli can occur during uncomplicated cerebral angiography. In the literature, microemboli were predominantly detected during contrast or saline flush injection and are seen as less commonly detected during catheter and wire manipulation [5]. In this patient, the spotty ischemic lesions on postoperative DWI likely resulted from cerebral emboli. DWI is very sensitive in detecting acute ischemic lesions immediately after onset [2]. This patient had another surgical risk factor; carotid ultrasonography revealed mild stenosis in the ipsilateral distal common carotid artery and the carotid bulb.

A multifactorial mechanism may be responsible for the development of optic disc ischemia in NAION. Whether optic disc ischemia results from local arteriosclerosis, thrombosis, embolization, generalized hypoperfusion, vasospasm, failure of autoregulation, or a combination of these processes is not known [6]. Although ophthalmologic findings with evidence of retinal embolism can be established in retinal artery occlusion, it is more difficult to define the relationship between NAION and embolism because the short posterior ciliary arteries (PCA) cannot be directly visualized. Instead, the embolic occlusion of the PCA was indirectly verified by segmental filling defects of the optic disc on FA. Embolic phenomena also can cause segmental papilledema and peripapillary retinal infarction seen on fundus photographs. The oval that forms the circle of Zinn can be divided into superior and inferior branches by the entry points of the medial and lateral short ciliary arteries forming it. This may be the anatomic basis for the altitudinal or quadrant visual field defect that characterizes this case.

## 4. Conclusions

Even without invasive endovascular procedures such as coil embolization near the ophthalmic artery’s origin, ophthalmic artery occlusion can occur due to procedure-associated microembolism. To detect newly developed visual field defects after TFCA, ophthalmic examinations and visual field tests should be performed before and immediately after TFCA, especially in high-risk patients who were either previously treated for intracranial aneurysm, have atherosclerotic cerebrovascular disease or a history of neuro-ophthalmological disorders. In addition, it can be helpful to prevent the risk of neuro-ophthalmic complications through the careful use of nephrotoxic drugs or contrast agents which can increase the risk of thromboembolic events, and the close follow-up of general conditions such as electrolyte balance during the angiography.

## Figures and Tables

**Figure 1 medicina-57-00567-f001:**
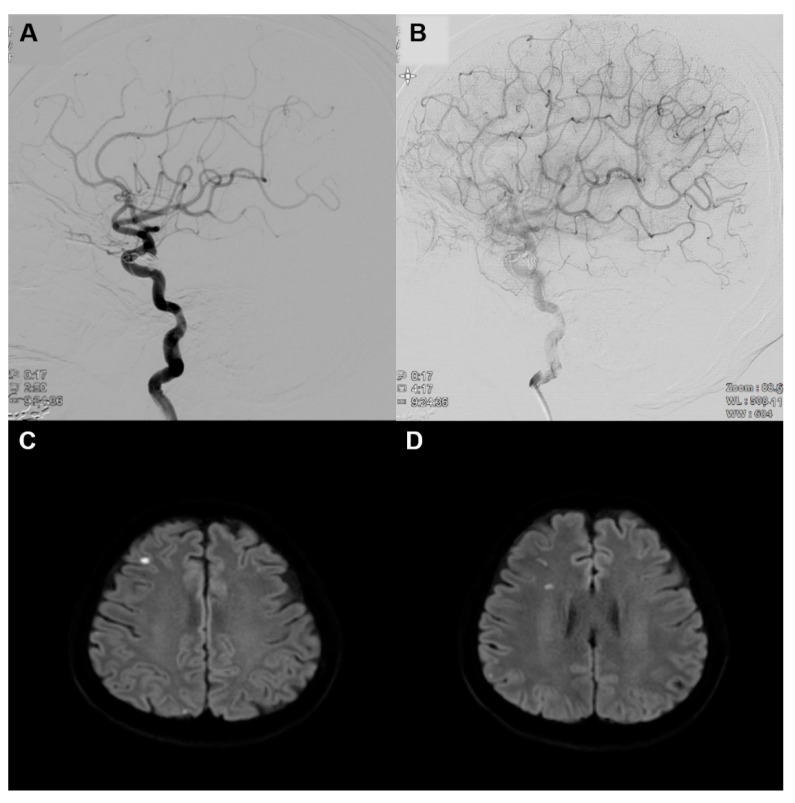
(**A**) Transfemoral cerebral angiography deconstrues the normal blood flow of the right ophthalmic artery. A previously clipped aneurysm is also noted on the right posterior communicating artery. (**B**) An intact choroidal blush is shown during the late arterial phase. (**C**,**D**) Diffusion-weighted MRI on the day following cerebral angiography revealed multiple focal scattered acute infarctions: right frontal, parietal, occipital, and temporal lobe.

**Figure 2 medicina-57-00567-f002:**
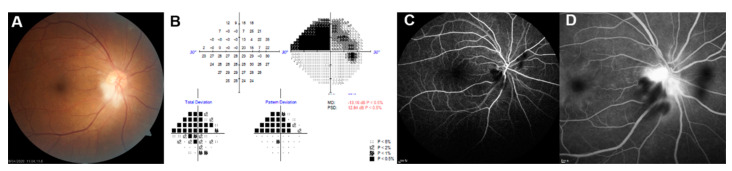
(**A**) Color fundus photo revealed inferior sectoral optic disc swelling and focal cotton wool spots in the peripapillary area of right eye. (**B**) Humphrey visual field test shows a superior altitudinal deficit. (**C**) Fluorescein angiography shows an early filling defect in the inferior segment of the optic nerve, and late leakage from optic disc (**D**).

**Figure 3 medicina-57-00567-f003:**
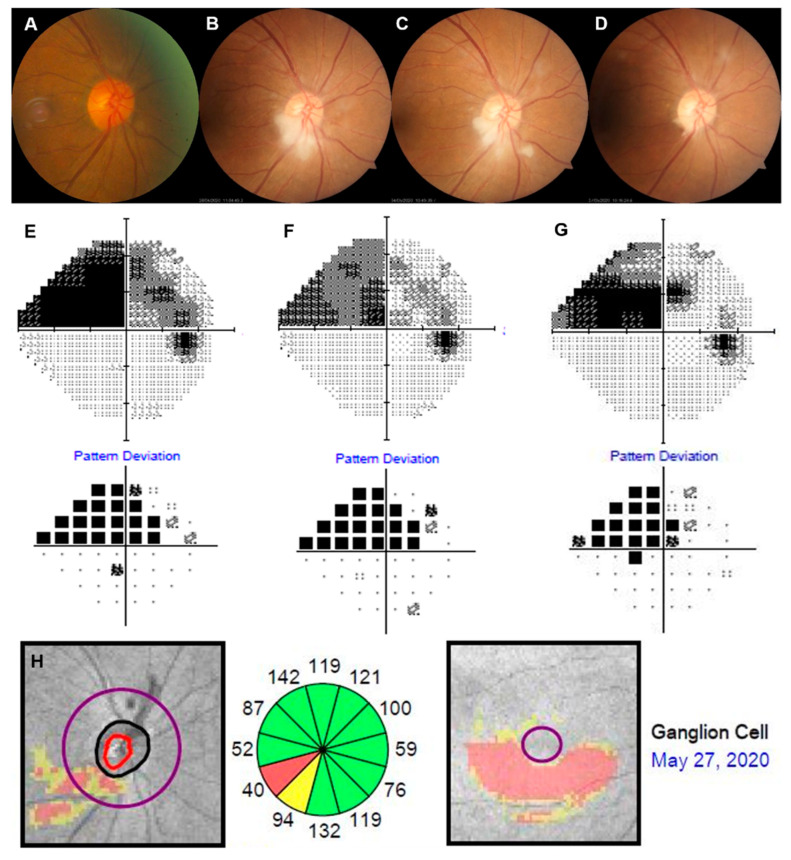
(**A**) Color optic disc photo 6 months before transfemoral cerebral angiography (TFCA) showed no definite abnormality. (**B**) Inferior optic disc edema and peripapillary retinal infarctions were initially found one day following TFCA. (**C**) Seven days after TFCA, steroid pulse therapy was completed, and optic disc edema was slightly resolved. (**D**) Thirty days after TFCA, optic disc edema and peripapillary lesions had almost disappeared. (**E**) The visual field of the right eye showed a superior altitudinal defect one day after TFCA. (**F**) Seven days after TFCA, minimal change in the superior defect in the visual field of the right eye. (**G**) Thirty days after TFCA, superior nasal field defect remained, with minimal improvement. (**H**) Inferior neuroretinal rim thinning and retinal nerve fiber layer (RNFL) defect in the inferior temporal quadrant were detected on spectral domain-optical coherence tomography.

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
