# Peer review of "Altitudinal Visual Field Defects Following Diagnostic Transfemoral Cerebral Angiography"

_medicina, 2021, doi:10.3390/medicina57060567_

Round 1

Reviewer 1 Report

The case report appears well organized and described in good English.

The setting of the work allows us to understand well how to quickly and adequately deal with the onset of a visual symptom such as altitude deficit. The choice of diagnostic tests is accurate and complete, the therapeutic approach is correct.

In the conclusions, as indicated in the notes, it would be preferable to read the ways to contain, as far as possible, the risk elements.

----------------------------------------------------------------------------------

Introduction

Line 27

The incidence of neurological complications following cerebral DSA has been reported.[1]

The incidence of neurological complications is reported in the discussion, but no mention is made of the purely ophthalmological ones, including the altitudinal defect described in the case report

It would be more didactic if this incidence were reported in the introduction, highlighting what are the ophthalmological complications and that percentage they have, compared to the other complications. In this way the reader immediately understands what the risk is to protect our patient from.

Line 31

A recent study found new visual defects in 33% of patients treated for ophthalmic segment internal carotid artery (ICA) aneurysms, although the majority of the population studied had undergone surgical clipping.

Probably, the authors want to highlight that if the TFCA it is not performed for vascular alterations in the ophthalmic area, there is practically no neurophthalmological damage.

In fact, the work cited concerns ophthalmic artery aneurysms and their treatment, not the TFCA analysis.

In such patients there are often pre-existing visual deficits due to damage to the optic nerve. The work highlights how surgical (clipping) or endoscopic (coiling) treatment in this area leads for various reasons to an increase in damage to the optic nerve.

Considering that the case report concerns the diagnostic practice of TFCA, the citation of the work in question could be better contextualized.

It would be more didactic to have references from the literature on the neuro-opthalmological complications of TFCA, clarifying that many patients already have a compromised thromboembolic and vascular picture and therefore at high risk. And this can cause damage through thromboembolic mechanisms even to fine structures such as the posterior ciliary arteries as well demonstrated in the presentation of the case report.

Discussion

Line 96

Until now, few cases of ocular complications have been reported, even after successful coil embolization of carotid-ophthalmic aneurysms.[3]

As already mentioned before, it would be nice to have the incidence of these few cases.

In another publication, cerebral angiography, performed on over 600 patients, was associated with a 1% overall incidence of neurologic deficit and a 0.5% incidence of persistent deficit. All complications occurred in patients presenting with a history of stroke / transient ischemic accident or carotid bruit, which may reflect the difficulty of performing angiography in this population at risk for atherosclerotic changes. But no ocular complications are reported (Heiserman JE. Neurologic Complications of Cerebral Angiography).

In the literature cited, complications rise to 2.63% out of 19826 cases, but even here the extent of the ophthalmological ones is not specified.

To understand the importance of ophthalmological complications and justify the conclusions of the work, it is necessary to understand how much they affect.

Conclusions

Line 124.

…ophthalmic artery occlusion can occur due to procedures associated microembolism. To detect newly developed visual field defects after TFCA, ophthalmic examinations and visual field tests should be performed before and immediately after TFCA, especially in high-risk patients who were either previously treated for intracranial aneurysm or have atherosclerotic cerebrovascular disease.

The authors report that the altitudinal defect is an isolated case following TFCA (and in fact the literature does not report anything in this sense). Even the review cited does not report anything about any neuro-ophthalmological complications. This warns the neuroradiologist and the ophthalmologist about the possibility, albeit remote, of having neurological complications also affecting the visual system.

The proposal to set up, before and after TFCA, a complex ophthalmological analysis, however logical and correct, appears to be difficult and expensive to carry out, precisely because of the rarity of the event.

All the more so due to the fact that the clinical outcomes are modest, despite the flawless and early diagnostic and clinical procedure operated, as reported by the authors.

The neuro-radiologist should instead evaluate the patient candidate for TFCA in advance and monitor him after the examination.

If neuro-ophthalmological disorders are present, these should be investigated before and after.

If the patient does not have visual problems, it is sufficient to leave an easy channel of communication open with the diagnostic staff for an analysis and early treatment.

Instead, it is essential to act preventively on the risk elements in order to reduce the Odds ratio: suspend the intake of nephrotoxic drugs when present at least 24-48 hours before the examination,  maintain adequate blood fluidity, promote adequate expansion of the circulating volume by administering intravenous electrolyte solutions a few hours before the examination, use non-iodized or low osmolar contrast media, have a specialist resuscitator during the procedure in the angiographic radiology room .

------------------------------------------------------------------------------------

Author Response

[Comment #1]

Line 27. The incidence of neurological complications following cerebral DSA has been reported.[1] The incidence of neurological complications is reported in the discussion, but no mention is made of the purely ophthalmological ones, including the altitudinal defect described in the case report.

It would be more didactic if this incidence were reported in the introduction, highlighting what are the ophthalmological complications and that percentage they have, compared to the other complications. In this way the reader immediately understands what the risk is to protect our patient from.

[Response #1]

I agreed to your comment, I added specific percentage of ophthalmic complication in the sentence.

"The incidence of neurological complications following cerebral DSA occurred in 2.63%, especially ophthalmic complications was 0.24%.”

[Comment #2]

Line 31

A recent study found new visual defects in 33% of patients treated for ophthalmic segment internal carotid artery (ICA) aneurysms, although the majority of the population studied had undergone surgical clipping.

Probably, the authors want to highlight that if the TFCA it is not performed for vascular alterations in the ophthalmic area, there is practically no neurophthalmological damage.

In fact, the work cited concerns ophthalmic artery aneurysms and their treatment, not the TFCA analysis.

In such patients there are often pre-existing visual deficits due to damage to the optic nerve. The work highlights how surgical (clipping) or endoscopic (coiling) treatment in this area leads for various reasons to an increase in damage to the optic nerve.

Considering that the case report concerns the diagnostic practice of TFCA, the citation of the work in question could be better contextualized.

It would be more didactic to have references from the literature on the neuro-opthalmological complications of TFCA, clarifying that many patients already have a compromised thromboembolic and vascular picture and therefore at high risk. And this can cause damage through thromboembolic mechanisms even to fine structures such as the posterior ciliary arteries as well demonstrated in the presentation of the case report.

[Response #2]

I definitely agreed your opinion, thank you for your comment. As I mentioned in response #1 after cerebral angiography, visual symptom was rarely observed, permanent visual symptom is much rare about 0.02%. Thus, exact percentage of classified visual field defect after TFCA or DSA was not yet reports, but more invasive procedure such as, coli embolization near ICA had much commonly observing neuro-ophthalmic complication than TFCA. Thus, I cited the incidence of invasive procedure. I added new citation to explain the low incidence of ophthalmic complication.

Previously reported that the incidence of permanent neuro-ophthalmological complications following cerebral DSA was much rare, about 0.02%. According to another paper, 0.2% of transient complications including visual field cut have been reported after cerebral an-giography using DSA. [1,3]

  1. Leffers AM, Wagner A, Neurologic complications of cerebral angiography; A retrospective study of complication rate and patient risk factors. Acta Radiologica. 2000; 40:204–210

[Comment #3]

Discussion

Line 96

Until now, few cases of ocular complications have been reported, even after successful coil embolization of carotid-ophthalmic aneurysms.[3]

As already mentioned before, it would be nice to have the incidence of these few cases.

In another publication, cerebral angiography, performed on over 600 patients, was associated with a 1% overall incidence of neurologic deficit and a 0.5% incidence of persistent deficit. All complications occurred in patients presenting with a history of stroke / transient ischemic accident or carotid bruit, which may reflect the difficulty of performing angiography in this population at risk for atherosclerotic changes. But no ocular complications are reported (Heiserman JE. Neurologic Complications of Cerebral Angiography).

In the literature cited, complications rise to 2.63% out of 19826 cases, but even here the extent of the ophthalmological ones is not specified.

To understand the importance of ophthalmological complications and justify the conclusions of the work, it is necessary to understand how much they affect.

[Response #3]

Thank you for your comment. On your advice, I added sentences including the incidence of neurologic defect, and permanent ophthalmologic defect. It improved the quality of my manuscript. I added specific percentage of ocular complication after successfule coiling embolization.

"Until now, 1.1% of ocular complications such as visual deficits have been reported, even after successful coil embolization of carotid ophthalmic aneurysms.”

[Comment #4]

Conclusions

Line 124.

…ophthalmic artery occlusion can occur due to procedures associated microembolism. To detect newly developed visual field defects after TFCA, ophthalmic examinations and visual field tests should be performed before and immediately after TFCA, especially in high-risk patients who were either previously treated for intracranial aneurysm or have atherosclerotic cerebrovascular disease.

The authors report that the altitudinal defect is an isolated case following TFCA (and in fact the literature does not report anything in this sense). Even the review cited does not report anything about any neuro-ophthalmological complications. This warns the neuroradiologist and the ophthalmologist about the possibility, albeit remote, of having neurological complications also affecting the visual system.

The proposal to set up, before and after TFCA, a complex ophthalmological analysis, however logical and correct, appears to be difficult and expensive to carry out, precisely because of the rarity of the event.

All the more so due to the fact that the clinical outcomes are modest, despite the flawless and early diagnostic and clinical procedure operated, as reported by the authors.

The neuro-radiologist should instead evaluate the patient candidate for TFCA in advance and monitor him after the examination.

If neuro-ophthalmological disorders are present, these should be investigated before and after.

If the patient does not have visual problems, it is sufficient to leave an easy channel of communication open with the diagnostic staff for an analysis and early treatment.

Instead, it is essential to act preventively on the risk elements in order to reduce the Odds ratio: suspend the intake of nephrotoxic drugs when present at least 24-48 hours before the examination, maintain adequate blood fluidity, promote adequate expansion of the circulating volume by administering intravenous electrolyte solutions a few hours before the examination, use non-iodized or low osmolar contrast media, have a specialist resuscitator during the procedure in the angiographic radiology room.

[Response #4]

  1. I agreed your mention. neuro-ophthalmological complication during the diagnostic TFCA was very rare cases. Also, permanent visual field defect much rare one, altitudinal visual field defect after TFCA, too. As you mentioned, complex ophthalmological test proposal only should have done with the patient who previously treated for intracranial aneurysm, have atherosclerotic cerebrovascular disease or history of neuro-ophthalmological disorders. Prior to TFCA, it should be accompanied by effort to reduce thromboembolic events on these patients. 124 was changed and added.

"To detect newly developed visual field defects after TFCA, ophthalmic examinations and visual field tests should be performed before and immediately after TFCA, especially in high-risk patients who were either previously treated for intracranial aneurysm, have atherosclerotic cerebrovascular disease or history of neuro-ophthalmological disorders. Finally, it is important and essential to prevent the risk of neuro-ophthalmic complications that careful use of nephrotoxic drugs or contrast agents which can increase the risk of thromboembolic events, and close follow up of general conditions such as electrolyte balance during the angiography.”

Reviewer 2 Report

" Nonarteritic anterior ischemic optic neuropathy is associated with cerebral small vessel disease"         

Kim MS et al. PLoS One. 2019 Nov 14;14(11):e0225322. doi: 10.1371

Accoring to this article, is it possible that the ischemic optic neuropahty could occur in association with cerebral small vessel disease irrespective of the TFCA because the fundus photo was taken the 6 months befor the TFCA in your case.

I wonder what you think about that.

Author Response

Dear Reviewer,

First of all, Thank you very much.

It is great honor for me to have mentioned from you.

[Comment #1]

" Nonarteritic anterior ischemic optic neuropathy is associated with cerebral small vessel disease"         

Kim MS et al. PLoS One. 2019 Nov 14;14(11):e0225322. doi: 10.1371

Accoring to this article, is it possible that the ischemic optic neuropahty could occur in association with cerebral small vessel disease irrespective of the TFCA because the fundus photo was taken the 6 months befor the TFCA in your case.

I wonder what you think about that.

[Response #1]

Cerebral SVD is highly associated with chronic degenerative changes leading to functional and cognitive disabilities, and have three subsets (WMH, CMB, and silent lacunar infarcts). 

Patients with NAION were more likely to show the chronic markers of cerebral SVD (such as WMH). 

In our case, brain CT angiography before TFCA, there was no focal lesion in the brain parenchyma. After TFCA, Symptom of visual field defect was occur next morning and focal scattered acute infarctions were observed on brain MRI angiography. 

Thus, we assumed ophthalmic artery occlusion occur due to procedures associated acute microembolic event and altitudinal or quadrant visual field defect may be the characteristics of this case.

I added the information of patient Brain CT angiography before TFCA, on paragraph "2. Case presentation".

"In brain computer tomography angiography before TFCA, there was no definite chronic degenerative changes or focal lesion in the brain parenchyma."

Thank you very much for your professional and detail comment, it will academically improving my case report.

This manuscript is a resubmission of an earlier submission. The following is a list of the peer review reports and author responses from that submission.